# Rett-causing mutations reveal two domains critical for MeCP2 function and for toxicity in *MECP2* duplication syndrome mice

Laura Dean Heckman[1,2], Maria H Chahrour[3], Huda Y Zoghbi[1,2,4]*

[1]Department of Molecular and Human Genetics, Baylor College of Medicine, Houston, United States; [2]Jan and Dan Duncan Neurological Research Institute at Texas Children's Hospital, Houston, United States; [3]Division of Genetics, Department of Medicine, Harvard Medical School, Boston, United States; [4]Howard Hughes Medical Institute, Baylor College of Medicine, Houston, United States

**Abstract** Loss of function of the X-linked gene encoding methyl-CpG binding protein 2 (MeCP2) causes the progressive neurological disorder Rett syndrome (RTT). Conversely, duplication or triplication of Xq28 causes an equally wide-ranging progressive neurological disorder, *MECP2* duplication syndrome, whose features overlap somewhat with RTT. To understand which MeCP2 functions cause toxicity in the duplication syndrome, we generated mouse models expressing endogenous *Mecp2* along with a RTT-causing mutation in either the methyl-CpG binding domain (MBD) or the transcriptional repression domain (TRD). We determined that both the MBD and TRD must function for doubling MeCP2 to be toxic. Mutating the MBD reproduces the null phenotype and expressing the TRD mutant produces milder RTT phenotypes, yet both mutations are harmless when expressed with endogenous *Mecp2*. Surprisingly, mutating the TRD is more detrimental than deleting the entire C-terminus, indicating a dominant-negative effect on MeCP2 function, likely due to the disruption of a basic cluster.

*For correspondence: hzoghbi@bcm.edu

**Reviewing editor**: Christian Rosenmund, Charité-Universitätsmedizin Berlin, Germany

## Introduction

Rett syndrome (RTT), the most common monogenic cause of intellectual disability in females, is a debilitating, progressive neurological disorder that is caused by mutations in the X-linked gene encoding methyl-CpG binding protein 2 (MeCP2) (*Amir et al., 1999*). After a 6- to 18-month period of normal development, head growth slows and affected girls lose acquired speech, dexterity, and social skills; they then develop characteristic hand stereotypies, respiratory dysrhythmias, seizures, and autistic-like features (*Hagberg et al., 1983*; *Lam et al., 2000*; *Klauck et al., 2002*; *Carney et al., 2003*). Interestingly, duplications and triplications spanning the *MECP2* region on Xq28 cause a similarly progressive neurological disorder called *MECP2* duplication syndrome, which has some features that overlap with RTT. Children with the duplication syndrome present with infantile hypotonia and develop severe intellectual disability, autistic-like features, recurrent respiratory infections, spasticity, seizures, and premature lethality (*del Gaudio et al., 2006*; *Friez et al., 2006*; *Lugtenberg et al., 2006*; *Meins et al., 2005*; *Van Esch et al., 2005*).

MeCP2 was first identified over 20 years ago as a transcriptional repressor that binds to methylated CpG dinucleotides (*Lewis et al., 1992*; *Wakefield et al., 1999*; *Free et al., 2001*). It binds DNA directly through its N-terminal methyl-CpG binding domain (MBD), whereas its C-terminal transcriptional repression domain (TRD) allows it to interact with corepressors such as Sin3a, HDAC1, and HDAC2

**eLife digest** Rett syndrome is a disorder that affects the development of the brain after birth. Infants with this condition develop as normal until they are 6–18 months old, when the development of their language and motor skills stops, or even regresses. Most cases of Rett syndrome are caused by mutations in a gene called *MECP2*.

If an individual mistakenly inherits an extra copy of the *MECP2* gene, it can cause another developmental disorder called *MECP2* duplication syndrome. This condition, which also affects the brain, gets worse over time and shares many features with Rett syndrome. The extra copy of the *MECP2* gene leads to the production of too much MeCP2 protein. However, how doubling the level of this protein causes the syndrome and, in particular, which parts of the protein are involved are unknown.

Previously, researchers engineered mice that expressed a copy of the human *MECP2* gene alongside their own version of the gene. These mice developed a condition similar to *MECP2* duplication syndrome and many of these mice suffered from seizures and died within their first year.

Heckman et al. have now engineered mice that also have an extra human *MECP2* gene but with one of two mutations that cause Rett syndrome in humans. Some mice had a mutation in a part of the MeCP2 protein that binds to DNA that is marked with small chemical tags called methyl groups. Other mice had a mutation in a domain of the protein that works to switch off genes. Heckman et al. found that mice with extra MeCP2 protein with either of these two mutations were as healthy as normal mice and showed none of the signs of *MECP2* duplication syndrome. This indicates that both of these domains must be intact for doubling the levels of the MeCP2 protein to be harmful. Furthermore, Heckman et al. discovered that the mutation in the part of MeCP2 that works to switch genes off also reduces the protein's ability to bind to DNA.

The next challenge is to understand the mechanism by which doubling the levels of this protein causes harm to the brain. Further work is also needed to uncover why having too much MeCP2 protein or none at all cause syndromes that share many features.

(*Nan et al., 1998*). More recent work has revealed that MeCP2 is expressed at higher levels than expected for classical site-specific transcriptional repressors: it binds as abundantly and widely throughout the genome as histone H1, which suggests MeCP2 might have additional functions in chromatin biology (*Nan et al., 1997*; *Skene et al., 2010*). Further complicating the picture of MeCP2 function are transcriptional studies in mouse brains as well as human embryonic stem cell-derived neurons, which have shown that most genes are actually downregulated in RTT models that lack MeCP2 (*Chahrour et al., 2008*; *Ben-Shachar et al., 2009*; *Li et al., 2013*). One proposal to explain this is that MeCP2 acts as a 'transcriptional noise dampener', such that loss of MeCP2 function results in the diversion of basal transcriptional machinery to repetitive elements, indirectly leading to global transcriptional downregulation (*Skene et al., 2010*). Additional transcriptional studies present a challenge for this hypothesis, however, as the same genes that are downregulated in the RTT models are upregulated in *MECP2* duplication syndrome mouse models with double the MeCP2 expression (*Chahrour et al., 2008*). How might doubling MeCP2 levels lead to the upregulation of the very same genes? It may be that, given the abundance of MeCP2 in the normal state, doubling the protein levels, as in the duplication syndrome, exhausts the supply of normal binding partners, thus diverting them from their normal functions, but this possibility has not been formally tested.

Because most RTT-causing mutations disrupt either the MBD or TRD, we decided to use a genetic approach to determine how these regions might be involved in mediating the toxicity observed in *MECP2* duplication syndrome models. Studying transgenic models with known RTT-causing missense mutations on a *Mecp2* null background would allow us to observe the effect of the mutation when it is the only allele present, as in RTT, whereas studying the effects of the transgene on a wild type (WT) background would enable us to study the importance of each domain to the neurotoxicity caused by *MECP2* duplication syndrome.

To explore the role of the MBD in RTT and *MECP2* duplication syndrome, we chose a RTT-causing point mutation that replaces an arginine with a glycine at residue 111 (R111G) (*Laccone et al., 2001*). R111 is critical for methyl-CpG binding in vitro: when it is mutated (R111G), it completely abolishes

binding to methyl-CpGs in vitro without affecting the structure of the MBD or the rest of the protein (*Free et al., 2001*; *Kudo et al., 2003*). We selected this mutation to determine whether MeCP2 retains any functions without a functional MBD, whether the MBD is required for the toxicity observed in duplication models, and whether doubling MeCP2 mediates toxicity by over-titrating its binding partners.

Much less is known about the TRD, but the importance of the C-terminus is underscored by the clustering of common RTT-causing mutations at the very end of the TRD (*Christodoulou et al., 2003*). In fact, the mutation of arginine to cysteine at residue 306 (R306C) is the second most common RTT-causing missense mutation. Although recent studies have shown that the R306C mutation abolishes interaction with the NCoR corepressor complex, a mouse model bearing a truncated form of MeCP2 that lacks the NCoR binding site has a milder phenotype than the published mice carrying the R306C mutation, indicating that R306C has other effects on the function of MeCP2 (*Baker et al., 2013*; *Ebert et al., 2013*; *Lyst et al., 2013*).

To study the functional consequences of these two mutations in vivo, we generated independent transgenic mouse models, each bearing one of these point mutations in the *MECP2* locus. We characterized the behavioral and molecular phenotypes of mice carrying only the mutant allele (and thus modeling RTT) and mice carrying the mutant allele in addition to a WT allele (modeling *MECP2* duplication syndrome) in an effort to determine how each mutation affects the function of the protein in the two diseases.

## Results

### Generation of transgenic mice carrying MeCP2 with the R111G or the R306C missense mutation

To generate a modified *MECP2* allele, we used a PAC (P1-derived artificial chromosome, PAC671D9) containing the entire human *MECP2* locus and its essential regulatory elements, but no other genes. This PAC clone was used to create the *MECP2* duplication mouse model and predict the human disease, and on a *Mecp2* null background it rescues loss-of-function phenotypes (*Collins et al., 2004*). We used recombineering to modify the locus to carry either the R111G or the R306C mutation (*Warming et al., 2005*). This PAC has reliably produced founders with protein levels and expression pattern similar to that of endogenous MeCP2 (*Collins et al., 2004*). We further modified the PAC to tag MeCP2 with enhanced green fluorescent protein (EGFP) at the C-terminus to better identify interactors through immunoprecipitation (IP) and visualize their localization using immunofluorescence (IF) without disrupting MeCP2 function. Arginine 111 (R111) is located in the middle of the MBD, whereas arginine 306 (R306) is in a cluster of basic amino acids at the end of the TRD (*Figure 1A*). After establishing transgenic lines for each mutation, we characterized the protein levels of two independent lines for each mutation by western blot and found that they expressed transgenic MeCP2 at levels similar to those of endogenous MeCP2, thus doubling the total MeCP2 level (*Figure 1B*). Because doubling MeCP2 consistently causes phenotypes similar to the human syndrome, we chose one line (Line 1) of each mutant for further characterization (*Collins et al., 2004*). Immunofluorescence using anti-MeCP2 and anti-GFP antibodies on midsagittal sections of the whole brain shows that the distribution pattern of transgenic MeCP2 parallels that of endogenous MeCP2 (*Figure 1C*, upper panel). This colocalization also holds true upon closer inspection of the cortex and cerebellum (*Figure 1C*, lower panel).

We crossed the transgenic males (*MECP2^R111G^-EGFP^Tg^* or *MECP2^R306C^-EGFP^Tg^*) to *Mecp2^+/−^* females (*Guy et al., 2001*). These crosses resulted in males that were wild type, null, modeled RTT, or modeled the duplication syndrome, specifically: (1) *Mecp2^+/y^* (hereafter WT), (2) *Mecp2^−/y^* (hereafter null), (3) *MECP2^R111G^-EGFP^Tg^; Mecp2^−/y^* or *MECP2^R306C^-EGFP^Tg^; Mecp2^−/y^* (hereafter R111G or R306C, respectively), and (4) *MECP2^R111G^-EGFP^Tg^; Mecp2^+/y^* or *MECP2^R306C^-EGFP^Tg^; Mecp2^+/y^* (hereafter R111G Tg or R306C Tg, respectively). Note that the R111G and R306C males express only the mutated version of MeCP2, whereas the R111G Tg and R306C Tg males express both endogenous *Mecp2* as well as the mutated *MECP2*.

### An intact MBD is critical for MeCP2 function and *MECP2* duplication syndrome phenotypes

*Mecp2* null mice started displaying symptoms around 4–6 weeks of age and had a median lifespan of 11 weeks (*Guy et al., 2001*) (*Figure 2A*, red line), and the R111G mice died as prematurely as the

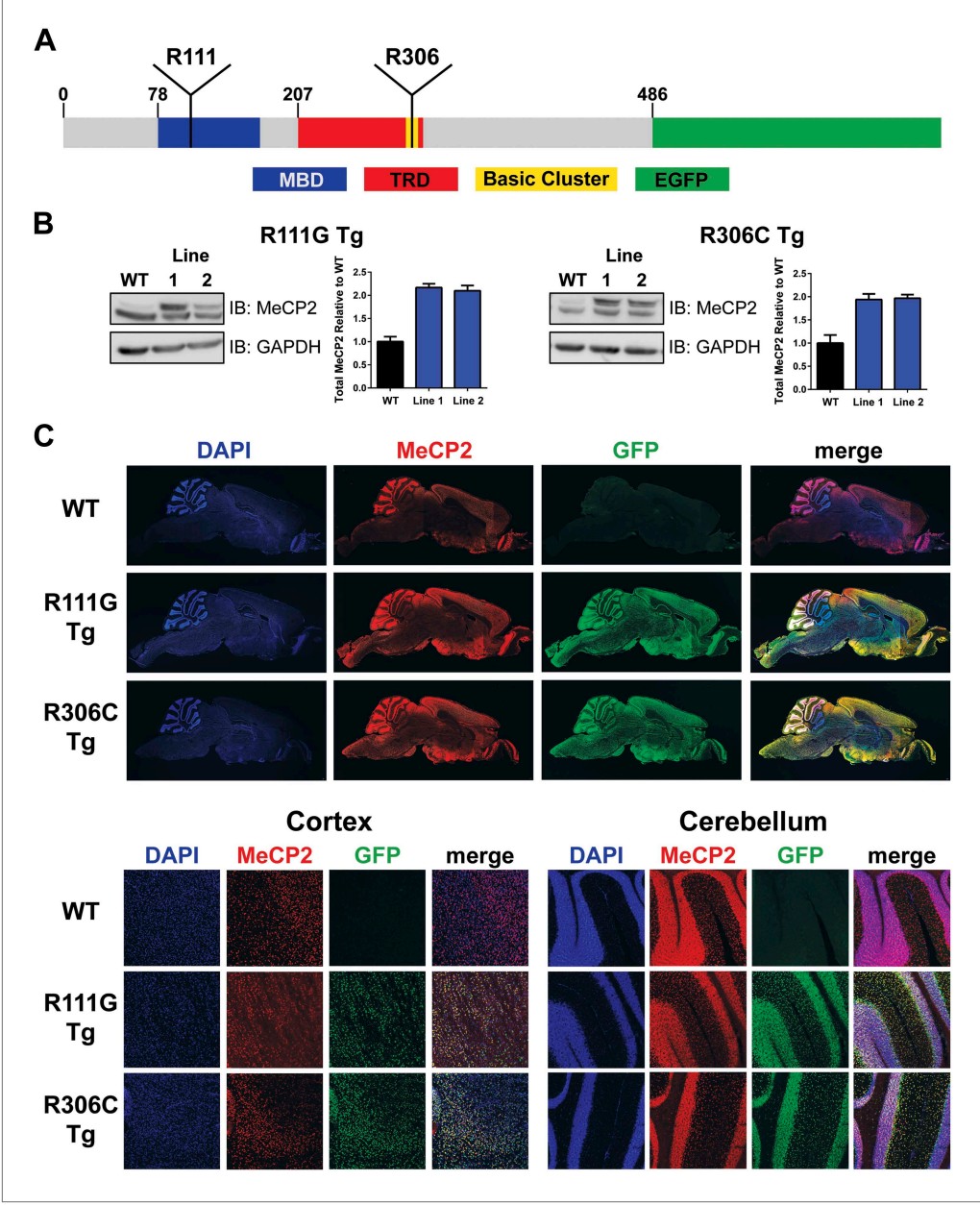

**Figure 1**. Generation of transgenic lines. (**A**) Schematic representation of MeCP2-EGFP fusion protein, showing the methyl-CpG binding domain (MBD, blue), transcriptional repression domain (TRD, red), a cluster of basic amino acids (yellow), and enhanced green fluorescent protein (EGFP, green). The location of the residues arginine 111 (R111) and arginine 306 (R306) are shown. (**B**) Western blot analyses of whole brain lysates show that two independent lines of MeCP2-R111G and MeCP2-R306C express the transgene at levels similar to endogenous MeCP2 as judged by the similar intensities of WT and transgenic bands using an anti-MeCP2 antibody. Quantification of total MeCP2 is shown to the right for each mutant. An anti-GAPDH antibody was used to detect GAPDH as a loading control. (**C**) Immunofluorescence (IF) of midsagittal brain sections (2 months, upper panel) shows that the expression pattern of the transgenes throughout the whole brain (anti-GFP antibody, green) parallels that of endogenous MeCP2 (anti-MeCP2 antibody, red). Colocalization is visible as yellow in the merged image, with 4',6-diamidino-2-phenylindole (DAPI, blue) as a general marker for the nucleus. Closer examination of the cortex and cerebellum (lower panel) mirrors this.

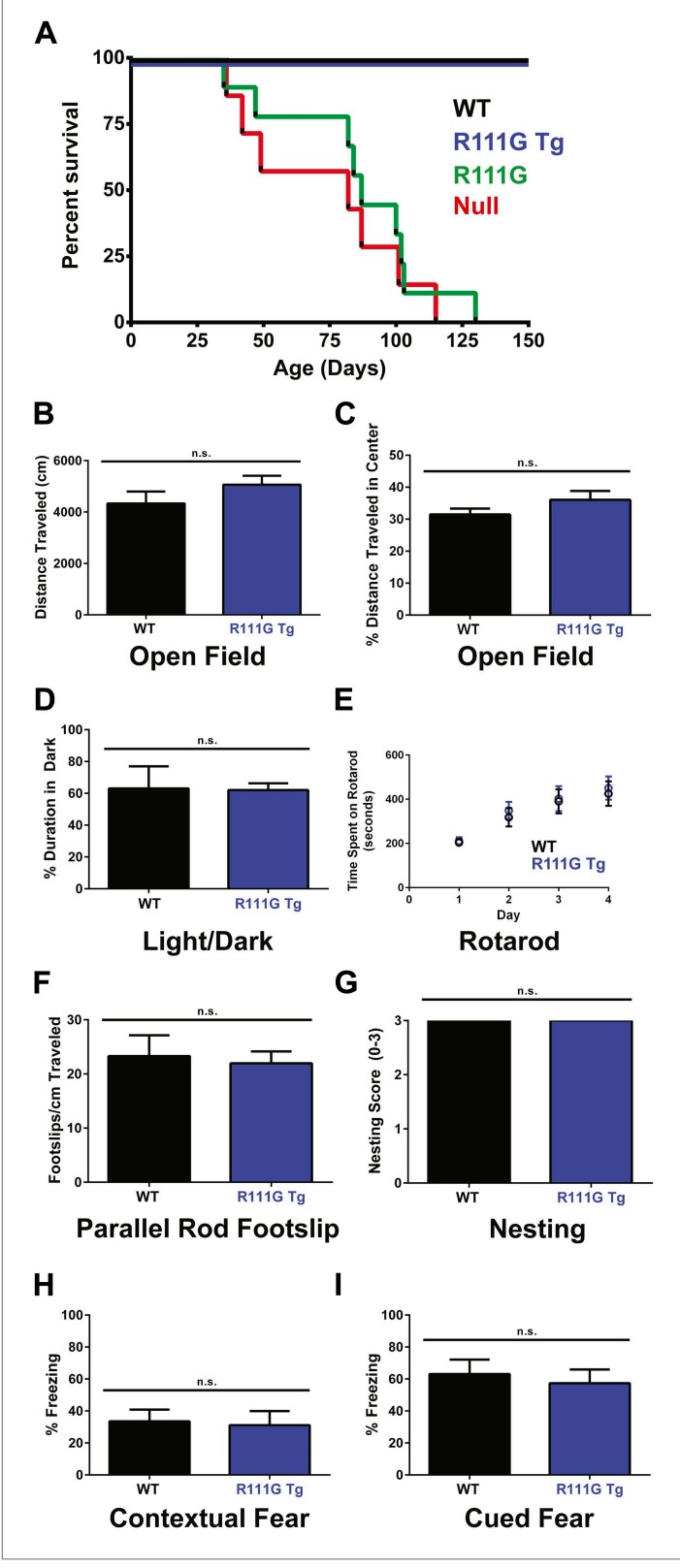

**Figure 2**. R111G mice phenocopy the null mice whereas R111G Tg mice are indistinguishable from WT mice. (**A**) Using a Kaplan–Meier survival curve, both WT (black, n = 11) and R111G Tg (blue, n = 10) have a normal
*Figure 2. Continued on next page*

null mice (*Figure 2A*, green line). This indicates that the R111G mutation disrupts a critical function of MeCP2, such that it becomes effectively a null allele.

Although the R111G Tg had a lifespan like the WT and appeared normal, we wanted to determine if they had any subtle behavioral deficits. Older mice expressing two functional copies of MeCP2 exhibit mild phenotypes, such as hypoactivity, reduced anxiety, and improved learning (*Collins et al., 2004*; *Na et al., 2012*), so we aged the WT and R111G Tg mice to 9 months before starting our battery of behavioral tests. We examined overall activity and anxiety using distance traveled in the open field assay (*Figure 2B*), the center-to-total ratio in the open field assay (*Figure 2C*), and time spent in the dark in the light/dark assay (*Figure 2D*). We also assayed motor coordination by their ability to stay on a rotating rod in the rotarod assay (*Figure 2E*), footslips per distance traveled in parallel rod footslip (*Figure 2F*), and nest building after 24 hr (*Figure 2G*). We assessed learning and memory with the contextual and cued conditioned fear test (*Figure 2H–I*). We found no difference between WT and R111G Tg mice in any of these behaviors (*Figure 2B–I*), indicating that an intact MBD is necessary to elicit the toxicity seen in the duplication syndrome model. These data also show that the phenotypes of the R111G mice (when the only MeCP2 allele is mutated) are exclusively caused by the mutation, and not insertional effects of the transgene, as the presence of a WT MeCP2 allele (in the R111G Tg mice) displays no phenotypes.

In vitro, MeCP2-R111G abolishes binding to methyl-CpGs (*Free et al., 2001*; *Kudo et al., 2003*). This had never been shown in vivo, however, so we used immunofluorescence (IF)

*Figure 2. Continued*

lifespan, whereas R111G mice (green, n = 10) phenocopy the premature lethality of the null mice (red, n = 11), with a median lifespan of 11 weeks. (**B**–**I**) Behavioral assays performed on WT (n = 7) and R111G Tg (n = 7) mice at 9 months of age show that R111G Tg mice are indistinguishable from WT mice in a variety of assays. The open field assay reveals that both lines travel the same distance (**B**) and have the same percentage of time traveled in the center (**C**), indicating that they are not anxious, which is confirmed by the percentage of time spent in the dark in the light/dark assay (**D**). Motor function and coordination were unchanged as measured by time spent on the rod (for four trials a day for four days) in rotarod (**E**) and footslips per centimeter traveled in parallel rod footslip (**F**). Purposeful paw movement is also normal, as assayed by nest building after 24 hr (**G**). R111G Tg mice also have no learning and memory deficits, as assayed by their freezing in the contextual (**H**) and cued (**I**) conditioned fear test. Data were analyzed by an ordinary one-way ANOVA followed by Tukey's multiple comparisons test. Results were plotted as the mean ± SEM. n.s. not significant.

and DAPI staining to visualize the localization of MeCP2 to heterochromatic foci, which are regions of highly compacted chromatin. The localization of a protein to these sites can serve as a proxy for methyl-CpG binding, and WT MeCP2 usually localizes there (*Figure 3A*, upper panel). We found that MeCP2-R111G was unable to localize to heterochromatic foci (*Figure 3A*, lower panel), suggesting it is unable to bind methyl-CpGs in vivo. To verify that abolishing methyl-CpG binding has no effect on the ability of MeCP2 to interact with binding partners, we performed in vivo immunopre-cipitation (IP) to determine whether MeCP2-R111G can interact with the Sin3a, HDAC1, and HDAC2 of the Sin3a corepressor complex and HDAC3 and Tbl1 of the NCoR corepressor complex (*Nan et al., 1998*; *Stancheva et al., 2003*). As expected, MeCP2-R111G retained the ability to interact with both corepressor complexes to the same degree as MeCP2-EGFP (*Figure 3B*). This demonstrates that the R111G mutation does not affect the ability of MeCP2 to bind corepressors, despite the fact that it disrupts binding to methyl-CpGs.

## R306C mice display milder phenotypes than *Mecp2* null mice

We next assessed the RTT-causing R306C mutation, located in the TRD. As with the R111G Tg mice, the R306C Tg mice were indistinguishable from their wild-type littermates by weight, lifespan, and brain size (*Figure 4A–C*). The R306C mice, however, displayed a milder phenotype than the *Mecp2* null mice. As early as 2 months of age, visual appraisal was sufficient to distinguish them: WT (and R306C Tg) mice had smooth coats, but the null mice appeared disheveled and the R306C mice were in-between. This milder phenotype was also reflected in the survival (*Figure 4A*) and weight (*Figure 4B*) curves: the null mice died early, with a median survival of 11 weeks (*Figure 4A*, red line) and were severely overweight (*Figure 4B*, red line), but the R306C mice had a median survival of 18 weeks (*Figure 4A*, green line) and were mildly overweight (*Figure 4B*, green line). Both R306C and null mice had smaller brains, mirroring the microcephaly seen in RTT patients (*Figure 4C*).

Localization of Alpha Thalassemia/Mental Retardation Syndrome X-Linked (ATRX) to heterochromatic foci has recently been demonstrated to be a cell-autonomous marker of disease progression in other RTT mouse models (*Baker et al., 2013*), so we evaluated ATRX localization using IF on comparable sections of the CA1, CA2, and CA3 regions of the hippocampus from 7-week-old mice (*Figure 4D*). The number of ATRX-positive foci per cell and focus intensity appeared to be the same in WT and R306C Tg (*Figure 4D*, upper two panels and quantification to the right). Null mice had the fewest ATRX-positive foci per cell and the lowest focus intensity, and R306C mice had an intermediate number of ATRX-positive foci per cell and focus intensity (*Figure 4D*, lower two panels and quantification to the right).

We next evaluated the animals for behavioral phenotypes. To fully characterize the R306C mice, we performed several behavioral tests at both 5 and 11 weeks of age. Most null mice die by 11 weeks, but they served as a point of comparison at the earlier timepoint; we were also able to compare later-manifesting phenotypes against the duplication model mice (R306C Tg). At both 5 and 11 weeks, the R306C Tg mice were indistinguishable from WT mice (*Figure 5*, comparing blue to black).

In contrast, at 5 weeks of age, R306C and null mice exhibited increased anxiety, as measured by increased time spent in the dark in the light/dark assay (*Figure 5A*). They displayed poor nest building (*Figure 5B*) and increased footslips per distance traveled in parallel rod footslip (*Figure 5C*), indicating deficits in purposeful paw movements and motor dysfunction. They also showed learning and memory

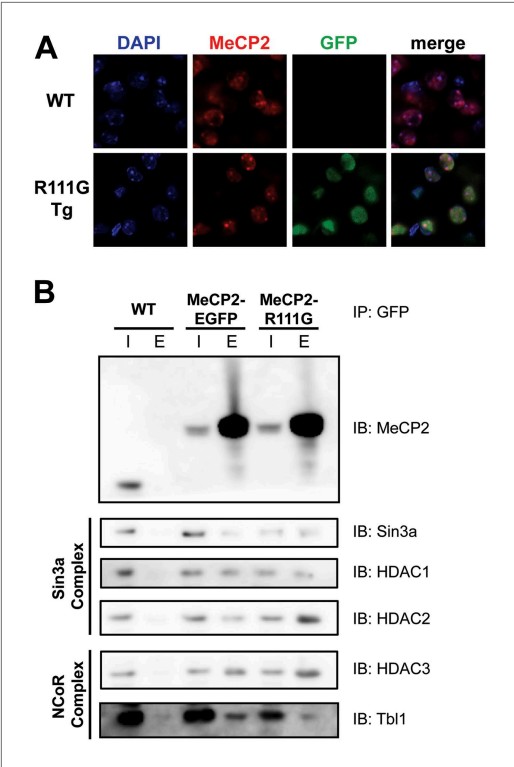

**Figure 3**. R111G abolishes MeCP2 binding to methyl-CpGs but does not affect its ability to interact with known binding partners. (**A**) MeCP2-R111G does not bind to methyl-CpGs in vivo. Immunofluorescence for endogenous MeCP2 (anti-MeCP2 antibody, red), the transgene (anti-GFP antibody, green), and heterochromatic foci (puncta within the nucleus visualized by DAPI, blue), shows that endogenous MeCP2 localizes to the heterochromatic foci and can therefore bind methyl-CpGs, whereas MeCP2-R111G is diffuse within the nucleus. (**B**) MeCP2-R111G retains the ability to bind known interactors in vivo. Immunoprecipitation (IP) with anti-GFP antibody was used to determine if MeCP2-R111G binds known interactors. Using WT mice lacking GFP as a negative control and MeCP2-EGFP as a positive control, we show that MeCP2-R111G retains binding to Sin3a, HDAC1, and HDAC2 of the Sin3a complex and HDAC3 and Tbl1 of the NCoR complex. I = input, E = eluate.

deficits as assayed by freezing in the contextual and cued conditioned fear assay (*Figure 5D–E*). Notably, in all tests, R306C mice had a less severe phenotype than the null mice.

By 11 weeks of age, R306C mice had become hypoactive as measured by distance traveled in the open field assay (*Figure 5F*). They continued to exhibit anxiety as demonstrated by their decreased time spent in the center in the open field assay (*Figure 5G*). Their motor dysfunction worsened (*Figure 5H*), as did their learning and memory deficits (*Figure 5I–J*). Additionally, a new 5-month old cohort showed that older R306C Tg mice have no motor coordination phenotype, as demonstrated by their ability to stay on a rotating rod in the rotarod assay similarly to WT (*Figure 5K*).

In summary, the R306C Tg mice displayed no phenotypes, indicating that the TRD and C-terminus must be intact to observe the toxic effects of doubling MeCP2. Mice carrying the RTT-causing R306C mutation recapitulated many of the phenotypes seen in RTT, including learning and memory deficits and motor dysfunction, indicating that this mouse model is a reliable model of RTT that can be used for future studies. As with the R111G Tg mice, the absence of phenotypes in the R306C Tg mice demonstrates that the phenotypes observed in the R306C mice are caused by the mutation and not insertional effects of the transgene.

## R306C reduces C-terminal binding to DNA, leading to reduced DNA occupancy in vivo

Because R306 is not located in or near the MBD, we did not predict that mutating it would affect binding to methyl-CpGs in vivo. We performed IF to visualize heterochromatic foci as before and confirmed that MeCP2-R306C retains the ability to bind methyl-CpGs in vivo in a fashion similar to endogenous MeCP2 (*Figure 6A*).

Since early reports suggested that the function of the C-terminus is to bind the corepressors Sin3a, HDAC1, and HDAC2 (*Nan et al., 1998*), we performed in vivo IP to assess the ability of MeCP2-R306C to interact with these corepressors. We found, to our surprise, that MeCP2-R306C bound

Sin3a, HDAC1, and HDAC2 similarly to WT EGFP-tagged MeCP2 (*Figure 6B*, upper half). We next tested components of the NCoR complex (*Stancheva et al., 2003*). In agreement with a recent study (*Lyst et al., 2013*), we found that R306C greatly reduces interaction with HDAC3 and Tbl1 of the NCoR corepressor complex (*Figure 6B*, lower half).

Although the loss of NCoR binding could play a role in the phenotypes we observe, we noticed a striking result when we compared the lifespans of previously published RTT models and the new R306C mice (*Figure 7A*). Mice with a truncating mutation, G273X, lack the entire region mapped to interact with NCoR, so MeCP2-G273X, like MeCP2-R306C, is unable to bind NCoR. However, the median lifespan of R306C mice (18 weeks, this study, *Lyst et al., 2013*) is much shorter than that of G273X mice (28 weeks, *Baker et al., 2013*). This led us to consider the potential differences between

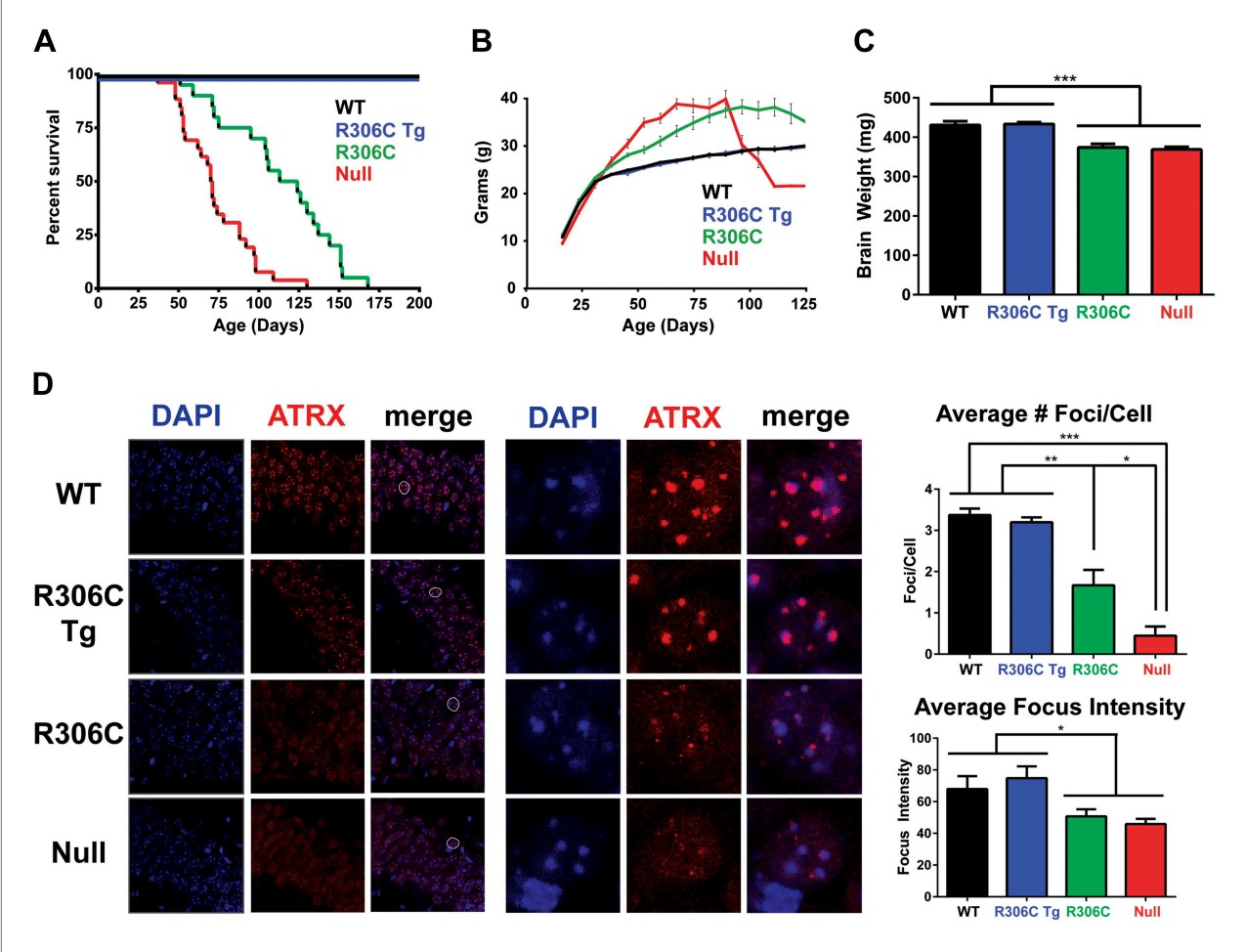

**Figure 4**. R306C mice have a milder phenotype than null mice. (**A**) A Kaplan–Meier survival curve shows that WT (black, n = 28) and R306C Tg (blue, n = 22) mice have a normal lifespan, while R306C (green, n = 21) mice have a shortened lifespan with 50% lethality at 18 weeks, but not as severely as the null mice (red, n = 26), with 50% survival at 11 weeks. (**B**) A weight curve shows that WT and R306C Tg mice have normal weights whereas R306C mice have a milder obesity phenotype. R306C mice are more overweight than WT and R306C Tg mice, but not as severely affected as the null mice. (**C**) R306C mice have smaller brains. At 7 weeks of age, WT and R306C Tg mice have similar brain sizes, whereas R306C and null mice have brains that are about 85% the normal weight. n = 7 per genotype. (**D**) ATRX localization to heterochromatic foci is used as a marker of disease severity. At 7 weeks of age, the number of ATRX-positive foci per cell (quantification to the upper right) and focus intensity (quantification to the lower right) is indistinguishable in WT and R306C Tg mice, while they are decreased in R306C, and even more decreased in null mice. The circled cell (white) in the merge of the left panel is shown at increased magnification in the center panel. Immunofluorescence was performed using DAPI (blue) and an anti-ATRX antibody (red). n = 3 per genotype. Data were analyzed by an ordinary one-way ANOVA followed by Tukey's multiple comparisons test. Results were plotted as the mean ± SEM. *p<0.05, **p<0.01, ***p<0.001.

the two alleles. Because MeCP2 shows affinity for specific sequences, we tested whether these mutants differed in their ability to bind these sequences by performing in vivo chromatin immunoprecipitation (ChIP) experiments on brain tissue (*Chahrour et al., 2008*; *Skene et al., 2010*; *Baker et al., 2013*). We found that MeCP2-R306C localized to the promoters of major satellite, L1 retrotransposons, somatostatin (*Sst*), afamin (*Afm*), corticotropin releasing hormone (*Crh*), and *Gapdh*, but bound to a lesser extent than MeCP2-G273X (*Figure 7B*).

Since the R306C mutation reduces binding to DNA in vivo, we aimed to determine the mechanism by which this occurs. Thus, we focused on the C-terminus of the protein and generated N-terminally GST-tagged C-terminal fragments of MeCP2 (amino acids 274–340) with and without the R306C mutation, which allowed us to uncover more subtle differences without the overwhelming DNA binding of the MBD. We performed electrophoretic mobility shift assays (EMSAs) to determine whether the C-terminal basic cluster flanking R306 binds DNA and further test whether the R306C mutation alters

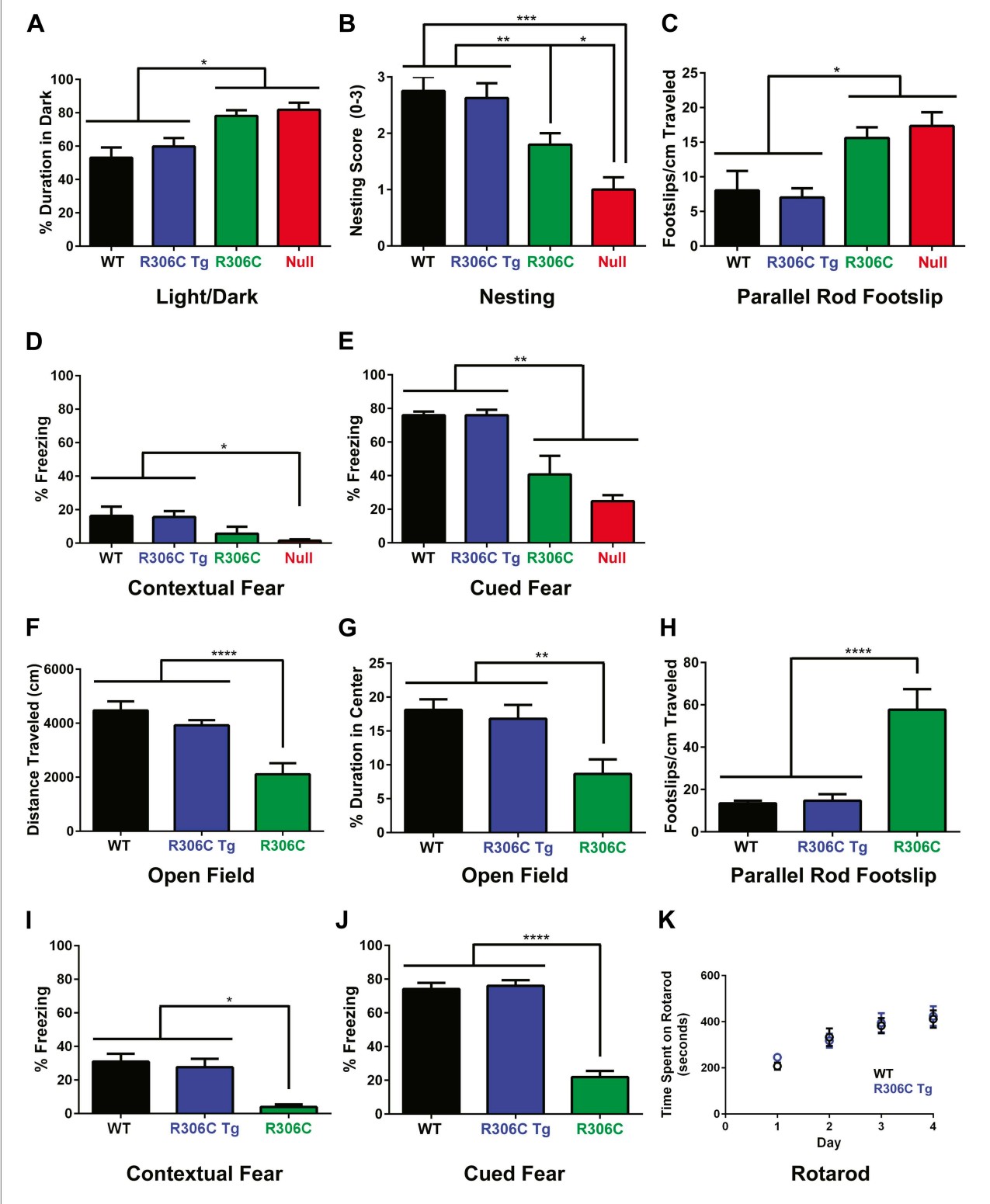

**Figure 5**. R306C mice recapitulate many RTT phenotypes. (**A–E**) Behavioral tests were performed at 5 weeks of age to examine the R306C phenotype. At this age, WT (black, n = 18) and R306C Tg (blue, n = 17) mice are indistinguishable. Like the null mice (red, n = 14), R306C mice (green, n = 16) have increased anxiety, as measured by increased time spent in the dark in the light/dark assay (**A**), purposeful paw movement deficits as measured by their ability to build nests (**B**), motor dysfunction as measured by increased footslips per cm traveled in parallel rod footslip (**C**), and learning and memory

*Figure 5. Continued on next page*

*Figure 5. Continued*

deficits in contextual (**D**) and cued (**E**) fear conditioning. (**F–J**) Behavioral tests were performed on a new cohort of mice at 11 weeks of age, when most of the null mice have succumbed to disease, to examine any changes in the observed phenotypes. WT (n = 17) and R306C Tg (n = 15) mice remain indistinguishable at this age. R306C (n = 15) mice were hypoactive in the open field assay (**F**) and continue to exhibit increased anxiety as measured by decreased time spent in the center in the open field assay (**G**). Additionally, their motor dysfunction and learning and memory deficits have both worsened (**H–J**). The motor coordination of older R306C Tg mice was tested with a new cohort of 5 month old WT (n = 12) and R306C Tg (n = 11) mice subjected to rotarod (four trials a day for 4 days), and no difference was observed in their time spent on the rod (**K**). Data were analyzed by an ordinary one-way ANOVA followed by Tukey's multiple comparisons test. Results were plotted as the mean ± SEM. *p<0.05, **p<0.01, ***p<0.001, ****p<0.0001.

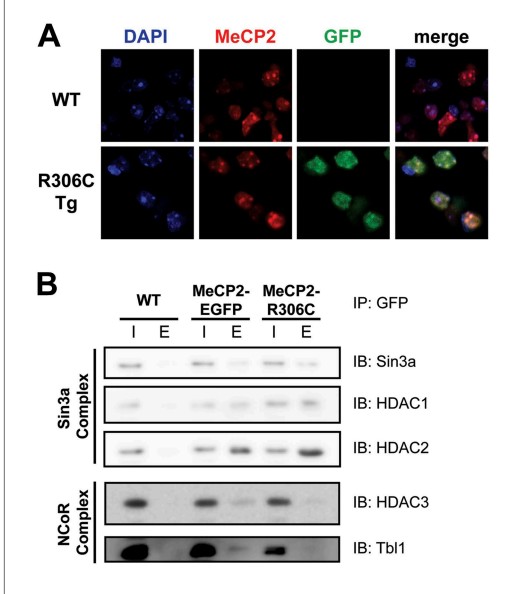

**Figure 6**. R306C does not affect methyl-CpG binding, but alters binding to a subset of known interactors. (**A**) MeCP2-R306C binds methyl-CpGs in vivo. Immunofluorescence for endogenous MeCP2 (anti-MeCP2 antibody, red), the transgene (anti-GFP antibody, green), and heterochromatic foci (puncta within the nucleus visualized by DAPI, blue) shows that both endogenous MeCP2 and MeCP2-R036C localize to the heterochromatic foci and can therefore bind methyl-CpGs. (**B**) MeCP2-R306C has altered binding to a subset of known interactors. Immunoprecipitation (IP) with anti-GFP antibody was used to determine if MeCP2-R306C binds known interactors. Using WT mice lacking GFP as a negative control and MeCP2-EGFP as a positive control, we show that MeCP2-R306C retains binding to Sin3a, HDAC1, and HDAC2 of the Sin3a complex, but not to HDAC3 and Tbl1 of the NCoR complex. I = input, E = eluate.

this binding. The WT MeCP2 fragment binds to the double-stranded DNA probe with increasing amount of protein, but the R306C mutation completely abolished the ability of the fragment to bind DNA (*Figure 7C*, two left panels). We further confirmed the importance of the basic cluster flanking R306 by performing EMSAs with two other RTT-causing point mutations: arginine to histidine at residue 306 (R306H) and lysine to glutamic acid at residue 304 (K304E), both of which remove a basic charge from the cluster (*Christodoulou et al., 2003*). These two mutations also abolish the ability of the fragment to bind DNA (*Figure 7C*, two right panels), thus emphasizing the importance of the basic cluster in C-terminal DNA binding. While these experiments were performed using a truncated MeCP2 peptide lacking the MBD (amino acids 274–340), we believe that the loss of this DNA binding is critical. In vivo ChIP analysis supports this notion, as despite the presence of the MBD and two functional AT-hooks in both MeCP2-G273X and MeCP2-R306C, the disruption of the basic cluster in MeCP2-R306C reduces binding to DNA even more than the complete absence of the cluster in MeCP2-G273X (*Figure 7B*). R306 therefore has an influence on DNA binding itself in addition to its role in NCoR complex binding, and having a defective C-terminus (R306C) is worse than having no C-terminus at all.

## Discussion

To gain insight into both RTT and *MECP2* duplication syndrome, we generated transgenic mouse models of MeCP2 carrying the R111G or the R306C missense mutation and compared them to the existing *Mecp2* null mice. Using the transgene on a *Mecp2* null background, we explored how the mutations disrupt normal function in RTT; using the transgene on a wild type background allowed us to explore whether the remaining functions of the mutant alleles can cause *MECP2* duplication syndrome.

Our investigation of the R111G mutation led to two important conclusions about the function of MeCP2. Since previous NMR studies demonstrated that this mutation abolishes binding to methyl-CpGs without affecting the structure of the MBD or the rest of the protein (*Free et al., 2001*), we used this mutation to assess the necessity of methyl-CpG binding in both RTT and *MECP2* duplication syndrome. If excessive MeCP2 disrupts neuronal function by diverting corepressors away from loci they

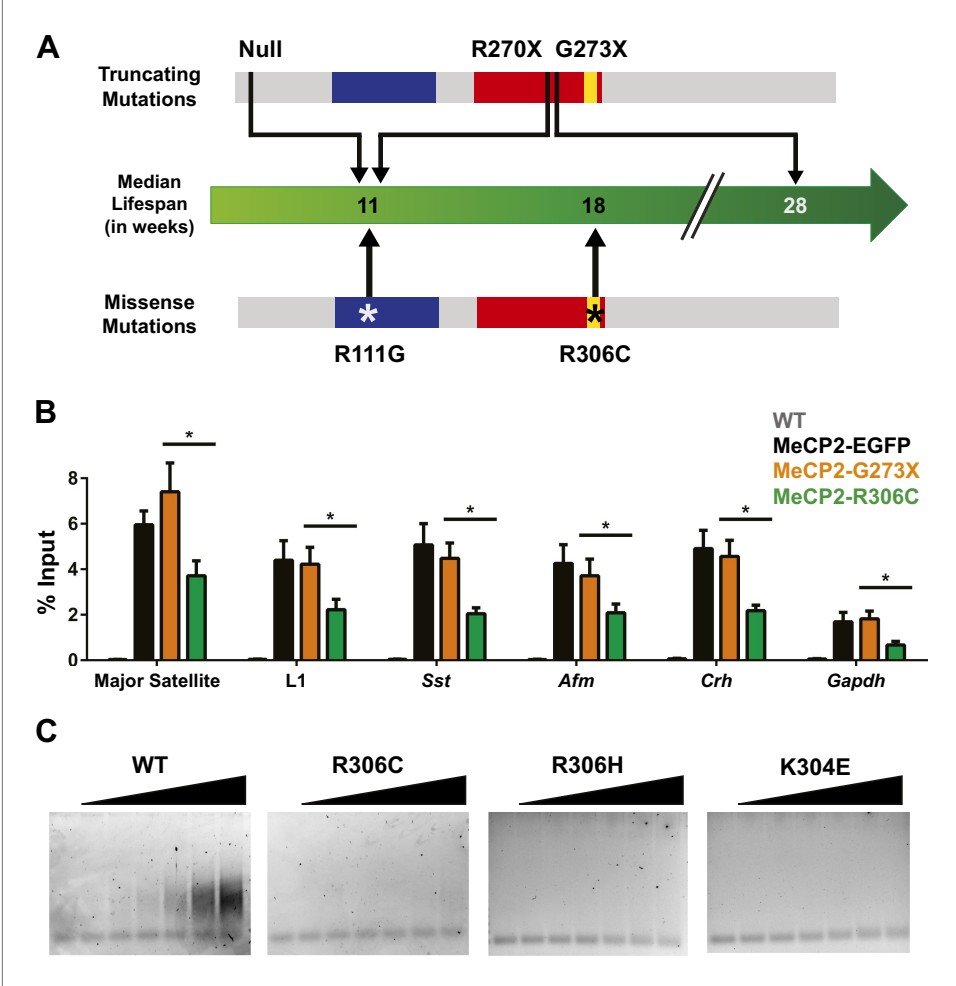

**Figure 7**. R306C decreases the affinity of the C-terminus of MeCP2 to DNA and reduces its DNA occupancy.
(**A**) An overview of the lifespans of RTT mouse models emphasizes the fact that the R306C missense mutation is more detrimental than an earlier truncation at G273X. Full-length MeCP2 (gray) with the MBD (blue), TRD (red), and basic cluster (yellow) is depicted. The top shows truncating mutations, while the bottom shows the missense mutations we generated. The median lifespans of mice carrying the mutations are shown in weeks.
(**B**) Chromatin immunoprecipitation (ChIP) using an anti-GFP antibody followed by quantitative PCR (qPCR) to determine enrichment of the following amplicons: Major Satellite repetitive sequences, L1 retrotransposons (L1), somatostatin (*Sst*), afamin (*Afm*), corticotropin releasing hormone (*Crh*), and *Gapdh* shows that MeCP2-R306C has decreased binding to sequences known to be bound by MeCP2 in vivo. Comparing MeCP2-R306C (green, n = 7) to MeCP2-G273X (orange, n = 7) shows that the differences observed are due to an additional effect of the C-terminus that is absent in the full-length MeCP2-EGFP (black, n = 4). WT mice with no GFP (gray, n = 3) were used as a negative control. (**C**) An electrophoretic mobility shift assay (EMSA) using increasing amounts of N-terminally GST-tagged C-terminal human MeCP2 recombinant protein (amino acids 274–340, from left to right: WT, R306C, R306H, K304E) shows that the WT C-terminal fragment of MeCP2 can bind DNA and that RTT-causing mutations altering the charge of basic residues in the fragment abolish this binding. Data were analyzed by an ordinary one-way ANOVA followed by Tukey's multiple comparisons test. Results were plotted as the mean ± SEM. *p<0.05.

normally repress, then abolishing methyl-CpG binding in an extra copy of MeCP2 that is still capable of interacting with corepressors (as we show in this study) would still produce the duplication phenotype in mice. Instead, the R111G Tg mice were indistinguishable from wild-type mice. Therefore, overabundant MeCP2 does not exert toxic effects because it diverts corepressors away from their normal binding sites: rather, trouble arises when more abundant MeCP2 occupies more methyl-CpGs. The centrality of methyl-CpG binding to MeCP2 function was underscored by the R111G mice, which

phenocopied the early lethality of the *Mecp2* null mice. All of this is consistent with studies asserting that lack of methyl-CpG binding leads to RTT (*Kudo et al., 2003*).

Like the R111G Tg mice, the R306C Tg mice were also indistinguishable from WT mice, demonstrating that a functional TRD is as necessary for the toxic effects of MeCP2 duplication as a functional MBD. The R306C mice, though displaying milder phenotypes than *Mecp2* null mice, recapitulated many phenotypes seen in RTT patients, such as increased anxiety, motor dysfunction, and learning and memory deficits (*Hagberg et al., 1983*; *Chahrour and Zoghbi, 2007*). This line is therefore a reliable mouse model for studying RTT and should be particularly useful since this is one of the most common RTT-causing mutations (*Christodoulou et al., 2003*).

The R306C mutation proved most useful for understanding the function of the less well-studied MeCP2 C-terminus. Because the G273X mouse model, which lacks the entire MeCP2 C-terminus, has a median survival of 28 weeks (*Baker et al., 2013*)—longer than both our R306C transgenic and the recently published R306C knock-in (median survival 18 weeks, this study, *Lyst et al., 2013*)—we hypothesized that the R306C allele must have an additional compromised function beyond its inability to bind NCoR through its TRD. Indeed, we found that MeCP2-R306C has reduced affinity for MeCP2 binding sequences in vivo, likely due to the disruption of a highly conserved basic cluster of amino acids (304-309) flanking R306, rendering it unable to bind DNA.

This study has added to our understanding of MeCP2 function by discovering that the basic cluster flanking R306 has an additional function in binding DNA and that this binding is abolished when basic residues in the cluster are mutated to neutral or acidic residues. Moreover, the R306C mutation is even more detrimental than lacking the entire basic cluster. This is reminiscent of a study documenting loss-of-function mutations in *SOB3*, a gene encoding an AT-hook-containing protein. The authors showed that an arginine to histidine disruption of the AT-hook produced a more severely impaired phenotype than a truncating mutation, indicating that the missense mutation has a dominant-negative effect on the function of the protein (*Street et al., 2008*). Mutation of R306 to either a cysteine or a histidine could have the same dominant-negative effect. Uncovering a role for this basic cluster adds the region around R306 to the MBD, TRD, and AT-hooks as domains known to be essential for MeCP2 function.

In sum, comparisons of mouse lines carrying individual disease-causing mutations provide great insights into the function of MeCP2 and the pathogenesis of RTT. In future studies, these and additional mouse models will help us to further investigate the role of MeCP2 and its DNA binding affinity in vivo.

## Materials and methods

### Generation of transgenic mice

The *galK* recombineering system (*Warming et al., 2005*) was used to modify a previously described PAC (P1-derived artificial chromosome, PAC671D9) containing the entire *MECP2* locus. In short, *galK*, flanked by 50 base pair (bp) homology arms to facilitate homologous recombination, was electroporated into induced SW102 cells containing the PAC to replace the residue of interest (R111 or R306). The presence of *galK* was selected for by growth at 30°C on minimal media plates containing galactose. Individual colonies were confirmed by streaking onto MacConkey plates followed by DNA sequencing. A single colony was chosen for the second round of electroporation to insert the desired mutation (AGG to GGG for R111G, CGC to TGC for R306C). The replacement of *galK* with the residue of interest was selected for by plating on minimal media plates containing 2-Deoxy-D-galactose. Individual colonies were screened for the correct insert size by PCR. DNA from a single colony was sequence verified, purified, and digested with NotI followed by pulsed-field gel electrophoresis to isolate the ~99 kb *MECP2*-continaing DNA from the backbone. To generate the transgenic mice, 1 ng/μl DNA was used for pronuclear injections into FVB single cell zygotes using standard procedures.

### Western blot analysis

Protein lysates were prepared from whole brains that were dounce homogenized in lysis buffer (20 mM Tris–HCl pH 8.0, 180 mM NaCl, 0.5% NP-40, 1 mM EDTA, Roche Complete Protease Inhibitor, Basel, Switzerland) followed by sonication and rotation for 30 min at 4°C. After centrifugation at 4°C to remove the insoluble fraction, the supernatant was mixed with 2X NuPAGE Sample Buffer and run on a NuPAGE 4–12% Bis-Tris gradient gel with MES Running Buffer (NuPAGE, Carlsbad, CA). Separated proteins were transferred to nitrocellulose membranes using the semi-dry method for 1 hr at 4°C.

The membranes were blocked with 10% milk in tris buffered saline with 2% Tween-20 (TBST), and incubated with primary antibody overnight at 4°C. After washing with TBST, the membranes were incubated with secondary antibody for 1 hr at room temperature followed by washing. HRP was detected using ECL detection kit (Pierce, Rockford, IL). Antibodies used were: rabbit antiserum raised against the N-terminus of MeCP2 (1:5,000; Zoghbi Lab, #0535), mouse anti-GAPDH 6C5 (1:20,000; Advanced Immunochemicals, 2-RGM2, Long Beach, CA), rabbit anti-Sin3a (1:2,000; Abcam, ab3479, Cambridge, England), rabbit anti-HDAC1 (1:2,000; Abcam, ab7028), rabbit anti-HDAC2 (1:2,000; Abcam, ab7029), rabbit anti-HDAC3 (1:500; Abcam, ab16047), and rabbit anti-Tbl1 (1:1000; Abcam, ab24548), donkey anti-rabbit HRP (1:20,000; GE Healthcare, NA934, Little Chalfont, United Kingdom), and donkey anti-mouse HRP (1:20,000; Jackson ImmunoResearch Labs, 715-035-150, West Grove, PA).

## Immunofluorescence

Immunofluorescence was performed as previously described with some modifications (*Chao et al., 2010*). Brains were dissected and cut midsagittally before drop-fixing in 4% paraformaldehyde (PFA, Sigma-Aldrich, St. Louis, MO) in phosphate buffered saline (PBS, Sigma–Aldrich) overnight at 4°C. Brains were cryoprotected in 30% sucrose for 1 day at 4°C and embedded in Optimum Cutting Temperature (O.C.T., Tissue-Tek, VWR, Radnor, PA). Free floating 30-μm midsagittal sections were cut on a Leica CM3050 S cryostat and stained for 24 hr with primary antibody, washed with TBST, and incubated for 24 hr with secondary antibody before mounting and imaging on a Leica TCS SP5 laser-scanning confocal microscope. Antibodies used were: rabbit anti-MeCP2 (1:1,000; Cell Signaling Technology, D4F3, Danvers, MA), rabbit anti-ATRX (1:100; Abcam, ab97508), and chicken anti-GFP (1:1,000; Abcam, ab13970), goat anti-rabbit (1:500; Alexa Fluor 555, Invitrogen, A-21428, Carlsbad, CA) and goat anti-chicken (1:500; Alexa Fluor 488, Invitrogen, A-11039). 4',6-diamidino-2-phenylindole (DAPI, Invitrogen) was used to stain DNA/nuclei.

## Mouse strains and genotyping

Transgenic R111G Tg and R306C Tg mice were maintained on a pure FVB background and genotyped using the following primers detecting GFP (Forward 5'-CAGCAGGACCATGTGATCGC-3', Reverse 5'-GTGAAGTTCGAGGTGCGACAC-3'). *Mecp2* null mice were backcrossed and maintained on a pure 129SvEvTac background and genotyped as previously described (*Guy et al., 2001*). Male transgenic mice were crossed to *Mecp2* heterozygous females (*Mecp2*$^{+/-}$) resulting in the 129SvEvTacxFVB F1 males that were used for all experiments.

All mouse studies were approved by the Institutional Animal Care and Use Committee for Baylor College of Medicine (IACUC Animal Welfare Assurance Number A3823-01), and animal housing, husbandry, and euthanasia were conducted under the guidelines of the Center for Comparative Medicine, Baylor College of Medicine (Protocol Number AN-1013).

## Behavioral assays

Mice were assayed at multiple time points by the following tests in the specified sequence so as not to interfere with the other tests: (1) open field assay, (2) light/dark assay, (3) nesting, (4) parallel rod footslip, and (5) contextual and cued conditioned fear, as previously described (*Chao et al., 2010*). Rotarod was performed on a new cohort of animals. Briefly, in the open field assay the movements of the mice in a 40 cm × 40 cm box, with 25 cm × 25 cm defined as the center, were tracked for 30 min and analyzed for total distance traveled and duration and distance traveled in the center using AccuScan Fusion software (Omnitech, Columbus, OH). In the light/dark assay, the movements of the mice in a 40 cm × 20 cm box (1/3 dark, 2/3 light) were tracked for 10 min and the time spent in the dark was analyzed using the AccuScan Fusion software. For nesting, mice were singly housed with a nestlet, and their ability to build a nest was assessed after 24 hr on a scale of 0–3 (0 = completely untouched, 3 = completely built nest). For parallel rod footslip, mice were placed in a 20 cm × 20 cm box and tracked for 10 min as they attempted to walk on a series of parallel rods. Footslips were detected by ANY-maze system (Stoelting, Wood Dale, IL), and footslips per centimeter traveled were analyzed. Conditioned fear was performed over 2 days using the Actimetrics chamber and FreezeFrame 3 system (Med Associates, St. Albans, VT): the mice received two 1 mA shocks accompanied by ~85 dB white noise on the first training day, and contextual fear was tested on the second day by placement in the same box with no white noise or shock. After a rest period of at least an hour, cued fear was tested by white noise with no shock in the same box altered by different visual and olfactory cues.

FreezeFrame 3 was used to analyze percent freezing during the white noise. For rotarod, mice were placed on an accelerating rotating rod (Ugo Basile, Varese, Italy) for four trials a day for four consecutive days for 10 min per trial, during which the rod accelerated from 4 to 40 rpm in the first 5 min. The time until the mouse fell (or rode around the rod twice in a row) was recorded and averaged for each day. Data were analyzed by an ordinary one-way ANOVA followed by Tukey's multiple comparisons test. Results were plotted as the mean ± the standard error of the mean (SEM).

## Immunoprecipitation

Whole brains were dissected and dounce homogenized in lysis buffer (20 mM Tris-HCl pH 8.0, 180 mM NaCl, 0.5% NP-40, 1 mM EDTA, Roche Complete Protease Inhibitor) followed by sonication and rotation for 30 min at 4°C for lysis. The insoluble fraction was removed by centrifugation, and the supernatant was rotated with camel anti-GFP beads (Chromotek-GFP-Trap beads, Allele Biotechnology, San Diego, CA) for 1 hr at 4°C. Beads were washed with lysis buffer before elution with 1X NuPAGE Sample Buffer by incubation at 70°C for 10 min. Inputs and eluates were subjected to western blot analysis.

## Chromatin immunoprecipitation-quantitative PCR

The chromatin immunoprecipitation (ChIP) was performed as previously described with some modifications (*Chahrour et al., 2008*). In short, whole brain tissue was extracted and minced before being crosslinked in 1% paraformaldehyde (PFA, Sigma-Aldrich) for 15 min at room temperature. Incubation with 125 mM glycine in PBS for 5 min at room temperature was used to quench the crosslinking reaction. The tissue was then dounce homogenized and lysed for 10 min on ice in cold lysis buffer (10 mM Tris-HCl pH 7.5, 10 mM NaCl, 3 mM $MgCl_2$, 0.5% NP-40, 1 mM PMSF, Roche Complete Protease Inhibitor, Ambion RNase cocktail, Carlsbad, CA), followed by washing with cold lysis buffer to isolate nuclei. Nuclei were collected by centrifugation and resuspended in micrococcal nuclease buffer (10 mM Tris-HCl pH 7.5, 10 mM NaCl, 3 mM $MgCl_2$, 1 mM $CaCl_2$, 4% NP-40, 1 mM PMSF, Roche Complete Protease Inhibitor) and sonicated. 75U MNase (Worthington Biochemical Corporation, Lakewood, NJ) was added to each sample and incubated at 37°C for exactly 5 min to digest chromatin to 100–300 bp fragments. The reaction was stopped by adding 2 mM EDTA, 1% SDS, and 200 mM NaCl. After an additional nuclear lysis by a second round of sonication followed by rotation at 4°C for 10 min, the lysate was cleared by centrifugation. The chromatin was confirmed to be sheared to 100–300 bp fragments using agarose gel electrophoresis before proceeding.

For the immunoprecipitation (IP), the chromatin was diluted 1:10 in ChIP dilution buffer (16.7 mM Tris–HCl pH 8.1, 167 mM NaCl, 0.01% SDS, 1.1% Triton X-100, 1.2 mM EDTA, 1 mg/ml BSA, 1 mM PMSF, Roche Complete Protease Inhibitor) and precleared with Protein A Dynabeads (Invitrogen) before incubation with rabbit anti-GFP antibody (3 µg, Abcam, ab6556) overnight at 4°C. Protein A Dynabeads were then added and incubated with the samples for 3 hr at 4°C. The beads were washed in low salt wash buffer (20 mM Tris–HCl pH 8.1, 150 mM NaCl, 0.1% SDS, 1% Triton X-100, 2 mM EDTA), high salt wash buffer (20 mM Tris-HCl pH 8.1, 500 mM NaCl, 0.1% SDS, 1% Triton X-100, 2 mM EDTA), LiCl wash buffer (10 mM Tris-HCl pH 8.1, 0.25 M LiCl, 1% NP-40, 1% deoxycholic acid, 1 mM EDTA), and twice in TE (10 mM Tris-HCl pH 7.4, 1 mM EDTA). The complexes were eluted twice in elution buffer (1% SDS, 100 mM $NaHCO_3$) for 15 min at room temperature. After reversal of the crosslinking by incubation at 65°C overnight, the DNA was treated with proteinase K (100 µg/ml, Bioline, London, United Kingdom) and Ambion RNase cocktail (2 µl per sample). DNA was isolated using phenol chloroform extraction followed by ethanol precipitation. The DNA pellet was resuspended in 30 µl TE buffer, and 1 µl was used for each individual quantitative PCR reaction.

Quantitative PCR (qPCR) reactions were performed in triplicate using iTaq SYBR Green Supermix (BioRad, Hercules, CA) on a CFX96 Real-Time System (BioRad) using a fast 2-step cycling program according to the manufacturer's instructions. The following primers were used to detect the promoters of Major Satellite: Forward 5'-CATCCACTTGACGACTTGAAAA-3' Reverse 5'-GAGGT CCTTCAGTGTGCATTT-3', L1 retrotransposon: Forward 5'-AGAAGAAACGGGAGACAGCA-3' Reverse 5'-CTGCCGTCTACTCCTCTTGG-3', somatostatin (*Sst*): Forward 5'-CATTGACAGGTACCCAACTGA-3' Reverse 5'-CAGCCACATAGGAGCACACTT-3', afamin (*Afm*): Forward 5'-AGACAGGCTGGCC TGAGAGTCA-3' Reverse 5'-TTCCAATGCACGCGTCTCACCC-3', corticotropin releasing hormone (*Crh*): Forward 5'-GTCACCAAGGAGGCGATACCTA-3' Reverse 5'-TAAATAATAGGGCCCTGCCAAG-3', *Gapdh*: Forward 5'-CCAGCTACTCGCGGCTTTACGG-3' Reverse 5'-CCTCCCGCCCTGCTTATCCAGT-3'. The threshold cycle (Ct) values were averaged for each sample and normalized to input. Data were

analyzed by an ordinary one-way ANOVA followed by Tukey's multiple comparisons test. Results were plotted as the mean ± SEM.

## Electrophoretic mobility shift assay

WT human MeCP2 and MeCP2-R306C (amino acids 274–340) were cloned in-frame to an N-terminal GST tag (GST-MeCP2 pGEX-5x3) using the In-Fusion EcoDry Cloning system (Clontech, Mountain View, CA), and C-terminal point mutations were generated using the QuikChange Multi Site-Directed Mutagenesis Kit (Agilent, Santa Clara, CA). Recombinant proteins were expressed in BL21(DE3) *E. coli* cells following 1 mM IPTG induction and purified using Glutathione Sepharose 4B beads (GE Healthcare) before elution into glutathione buffer (50 mM Tris–HCl pH 8.0, 10 mM glutathione) and dialyzed into PBS overnight at 4°C. Proteins were stored at −80°C.

The electrophoretic mobility shift assay (EMSA) was performed as previously described with some modifications (*Baker et al., 2013*). In short, increasing amounts of protein were combined with 0.1 mg/ml BSA, 66.7 nM DNA probe, and EMSA buffer (10 mM Tris–HCl pH 7.4, 50 mM KCl, 0.5 mM MgCl$_2$, 0.1 mM EDTA, 5% glycerol) and incubated at room temperature for 30 min before running on a 1% agarose gel in 1X TAE at 4°C. The strands of the 64-bp probe were annealed by boiling both strands for 5 min and cooling down to room temperature over the course of 3 hr. The sequence was as follows: Forward 5'-GGACTCCAGGTCCAGGACCGCGTTTTTCGCGCGCACGGCGCGGGAGGTCC AGCTGTCCACCTCC-3', Reverse 5'-GGAGGTGGACAGCTGGACCTCCCGCGCCGTGCGCGCGAAA AACGCGGTCCTGGACCTGGAGTCC-3'. Gels were post-stained with ethidium bromide and imaged under ultraviolet light to detect DNA.

## Acknowledgements

We would like to thank members of the Zoghbi lab, especially Steven Baker, and Vicky Brandt for discussions and critical input on the manuscript.

This work was supported by grants from the National Institutes of Health NINDS R01NS057819 (HYZ), the Mouse Neurobehavior Core of the Baylor College of Medicine Intellectual and Developmental Disabilities Research Center P30HD024064 (HYZ), NINDS F31NS077621 (LDH), and NIGMS T32GM008307 (LDH). HYZ is a Howard Hughes Medical Institute investigator.

## Additional information

### Competing interests

HYZ: Senior editor, *eLife*. The other authors declare that no competing interests exist.

### Funding

| Funder | Grant reference number | Author |
| --- | --- | --- |
| National Institute of Neurological Disorders and Stroke | R01NS057819 | Huda Y Zoghbi |
| Mouse Neurobehavior Core of the Baylor College of Medicine Intellectual and Developmental Disabilities Research Center | P30HD024064 | Huda Y Zoghbi |
| National Institute of Neurological Disorders and Stroke | F31NS077621 | Laura Dean Heckman |
| National Institute of General Medical Sciences | T32GM008307 | Laura Dean Heckman |
| Howard Hughes Medical Institute | | Huda Y Zoghbi |

The funders had no role in study design, data collection and interpretation, or the decision to submit the work for publication.

### Author contributions

LDH, Conception and design, Acquisition of data, Analysis and interpretation of data, Drafting or revising the article, Contributed unpublished essential data or reagents; MHC, Made wild type MeCP2-EGFP mouse, Acquisition of data; HYZ, Conception and design, Analysis and interpretation of data, Drafting or revising the article

## Ethics

Animal experimentation: All mouse studies were approved by the Institutional Animal Care and Use Committee for Baylor College of Medicine (IACUC Animal Welfare Assurance Number A3823-01), and animal housing, husbandry, and euthanasia were conducted under the guidelines of the Center for Comparative Medicine, Baylor College of Medicine (Protocol Number AN-1013).

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
