## [Decision Letter]

Thank you for sending your work entitled “Rett-causing mutations reveal domains critical for MeCP2 function and for toxicity in *MECP2* duplication syndrome mice” for consideration at *eLife*. Your article has been favorably evaluated by a Senior editor and 2 reviewers, one of whom is a member of our Board of Reviewing Editors.

The Reviewing editor and the other reviewer discussed their comments before we reached this decision, and the Reviewing editor has assembled the following comments to help you prepare a revised submission.

In this article, the authors have generated transgenic mouse models of MeCP2 carrying the R111G or R306C RTT missense mutations (on a null as well as WT background) to investigate the functionality of the methyl CpG- binding domain (MBD) and transcriptional repression domain (TRD). They have characterized the mouse models using behavioral and biochemical techniques and the basic findings include the following: (i) MeCP2-R111G behaves similar to null mice and does not bind methyl-CpGs in vivo, although interaction with Sin3a/NCoR corepressor complexes remains unaffected; (ii) R306C mice show milder phenotypes than MeCP2 null mice and retains ability to bind Methyl-CpG, Sin3A, HDAC1 and HDAC2 but displays reduced interaction with HDAC3 and TbI1 of NCoR corepressor complex; (iii) the C-terminus basic cluster (amino acids 274-340) might have an additional role in DNA binding since R306C reduces binding to DNA in vivo, also indicating a dominant-negative effect on MeCP2 function. The experiments are well done and contribute to our understanding of MeCP2 function in MECP2 duplication syndrome. Generally, the behavioral characterization and various biochemical techniques used in the paper help support the findings and confirm previous studies and provide important insights in the differential disease phenotype of Rett patients with mutations in Mecp2.

1) The detailed study of R306C adds strength to the paper and shows evidence for the C-terminus basic cluster to have a possible DNA binding role. However, some additional follow-up experiments are desirable but not required to confirm results shown in Figure 7. For example, (i) comparison to other MeCP2 mutations close to the basic cluster would be useful (ii) proof that DNA binding of the full length MeCP2-R306C is also impaired.

2) Two of the most pronounced phenotypes reported in MeCP2 overexpression in mice are context dependent fear conditioning and motor coordination impairments on the rotating rod ([9]; Na et al., 2013, which should also be cited). It would strengthen the findings that mutations in either functional domain do not produce phenotypes when expressed with endogenous MeCP2, if the authors examined the R111G-Tg and the R306-Tg mice in these behavioral paradigms. 

3) The authors demonstrated no behavioral effects of R111G-Tg and R306C-Tg mice in comparison to wild type animals. The authors should address the extent of interaction of R306C-Tg and R111G-Tg (those expressing mutant MeCP2 with endogenous MeCP2) with components of the Sin3a and NCoR repressor complexes. In the R306C-Tg and R111G-Tg mice, does MeCP2 interact with the corepressors equally to what is observed in the wild type animals? 

4) The immunofluorescence staining in Figure 6 demonstrates nuclear punctuated staining of R306C, indicative of DNA binding to heterochromatin. However, the EMSA data of R306C in Figure 7 shows its inability to bind DNA. These two pieces of evidences are somewhat conflicting so the authors should comment on this point.

---

## [Author Response]

*1) The detailed study of R306C adds strength to the paper and shows evidence for the C-terminus basic cluster to have a possible DNA binding role. However, some additional follow-up experiments are desirable but not required to confirm results shown in*
Figure 7*. For example, (i) comparison to other MeCP2 mutations close to the basic cluster would be useful (ii) proof that DNA binding of the full length MeCP2-R306C is also impaired*.

We have performed additional EMSA experiments to address (i) if other mutations in the basic cluster behave similarly and (ii) if the DNA binding of full-length MeCP2-R306C is impaired. We chose to pursue two particularly interesting RTT-causing mutations within the basic cluster: arginine to histidine at residue 306 (R306H) and lysine to glutamic acid at residue 304 (K304E). Both of these mutations change a positively charged amino acid to a relatively uncharged one (histidine) or negatively charged one (glutamic acid), whereas the other mutations in the cluster retain the positive charge. Additionally, within the cluster, R306H is the second most common RTT-causing missense mutation after R306C, indicating that this residue could be of particular importance. Using the same 274-340 amino acid fragment of MeCP2, we found that neither R306H nor K304E bind DNA, thus supporting our argument that the basic cluster flanking R306 is important for DNA binding. We included these findings in the revised Figure 7.

In the context of the full-length protein, we examined the binding of WT MeCP2 and MeCP2-R306C and found that both bind DNA with equal affinity in this *in vitro* assay. However, this is expected, as the primary mechanism of MeCP2 binding to DNA is via the methyl-CpG binding domain (MBD), thus making subtle differences in binding affinities extremely difficult to detect *in vitro*. Stronger proof that there is indeed a difference in the DNA binding of full-length MeCP2-R306C comes from the *in vivo* ChIP experiments. In Figure 7, it is clear that MeCP2-R306C-EGFP binding is decreased *in vivo* when compared to the wild type MeCP2-EGFP. Since the MBD is intact in both proteins and the only difference is the R306C mutation, it is safe to conclude that the mutation affects DNA binding in the context of the full-length protein. We made this point more clearly in the revised text.

*2) Two of the most pronounced phenotypes reported in MeCP2 overexpression in mice are context dependent fear conditioning and motor coordination impairments on the rotating rod (*[9]*; Na et al., 2013, which should also be cited). It would strengthen the findings that mutations in either functional domain do not produce phenotypes when expressed with endogenous MeCP2, if the authors examined the R111G-Tg and the R306-Tg mice in these behavioral paradigms*.* *

We thank the reviewers for the suggestion. We added the missing citation and performed the contextual conditioned fear and rotarod assays in the R111G Tg (new Figure 2) and R306C Tg (new Figure 5) mice. We found that there was no difference in these two lines when compared to WT littermates, thus strengthening our conclusion that both functional domains are necessary to see *MECP2* duplication phenotypes.

3) The authors demonstrated no behavioral effects of R111G-Tg and R306C-Tg mice in comparison to wild type animals. The authors should address the extent of interaction of R306C-Tg and R111G-Tg (those expressing mutant MeCP2 with endogenous MeCP2) with components of the Sin3a and NCoR repressor complexes. In the R306C-Tg and R111G-Tg mice, does MeCP2 interact with the corepressors equally to what is observed in the wild type animals? 

While we agree with the reviewers that this is an interesting question; the challenge of proving that endogenous MeCP2 interacts with the corepressor complexes in the transgenic animals lies in lack of good commercial MeCP2 antisera. The MeCP2 antibodies are not efficient, hence the reason we added an EGFP tag to our mutant protein. To pull down both WT and mutant MeCP2, it would be necessary to combine the MeCP2-EGFP allele with either the MeCP2-R111G or the MeCP2-R306C allele, on the *Mecp2* null background in one mouse. This will take 8 months or so and yet will not impact conclusions as explained in the next paragraph.

We can safely conclude that the WT endogenous MeCP2 is functional as it rescues all the R111G or R306C loss of function phenotypes: the transgenic mice behave exactly like WT animals indicating the endogenous protein is fully functional. Lastly our conclusion that both functional domains are necessary to cause phenotypes in the duplication mice would not be impacted by the outcome of endogenous MeCP2 IP. For example, MeCP2-R111G interacts with both corepressor complexes similarly to WT MeCP2 (Figure 3), so we would expect the two versions of the protein to interact with corepressors to the same extent in the duplication context. Yet we already know that duplication mice present with phenotypes whereas the R111G Tg mice do not. This indicates that interaction with the corepressors is not sufficient to cause duplication phenotype and requires the methyl-CpG binding domain (MBD).

*4) The immunofluorescence staining in*
Figure 6
*demonstrates nuclear punctuated staining of R306C, indicative of DNA binding to heterochromatin. However, the EMSA data of R306C in*
Figure 7
*shows its inability to bind DNA. These two pieces of evidences are somewhat conflicting so the authors should comment on this point*.

MeCP2 predominantly binds DNA via its methyl-CpG binding domain (MBD); what we show in our study is that the basic cluster can also bind DNA. To prove the latter point one has to use a fragment (amino acids 274-340 of MeCP2, thus lacking the MBD) in the *in vitro* EMSA assay. In contrast, the immunofluorescence in Figure 6 depicts full-length MeCP2-R306C localizing to heterochromatin *in vivo*. We chose to use the C-terminal fragment for the EMSA assay to uncover more subtle binding differences which would otherwise be obscured by the strong MBD binding. The *in vivo* ChIP assay, which is more quantitative than IF, shows that, on the whole, R306C binds a little less than WT protein in spite of its intact MBD, consistent with our conclusion that the basic cluster contributes some binding. We have clarified the discussion of these data in the text.